# Potential Prognostic Markers for Relapsed/Refractory vs. Responsive Acute Myeloid Leukemia

**DOI:** 10.3390/cancers14112752

**Published:** 2022-06-01

**Authors:** Aida Vitkevičienė, Giedrė Skliutė, Andrius Žučenka, Veronika Borutinskaitė, Rūta Navakauskienė

**Affiliations:** 1Department of Molecular Cell Biology, Institute of Biochemistry, Life Sciences Center, Vilnius University, Sauletekio av. 7, LT-01257 Vilnius, Lithuania; aida.vitkeviciene@gmail.com (A.V.); giedre.skliute@bchi.stud.vu.lt (G.S.); veronika.borutinskaite@bchi.vu.lt (V.B.); 2Hematology, Oncology and Transfusion Medicine Centre, Vilnius University Hospital Santaros Klinikos, Santariskiu str. 2, LT-08661 Vilnius, Lithuania; andrius.zucenka@santa.lt

**Keywords:** acute myeloid leukemia (AML), prognostic markers, epigenetic regulation

## Abstract

**Simple Summary:**

Acute myeloid leukemia (AML) is the most common blood cancer in the elderly, which progresses rapidly and is often fatal. The prognosis for AML remains poor in most older patients: only about 15% of patients over 60 years of age can recover. Our aim is to determine new potential AML clinical treatment prognosis markers. We analyzed certain genes, proteins and the epigenome profile in therapy-resistant and responsive AML patients at diagnosis stage and after clinical treatment. We determined that *MYC, WT1, IDH1, CDKN1A, HDAC2, TET1, KAT6A* and *GATAD2A* gene expression changes might characterize refractory AML. Therefore, these genes could have an impact for AML prognosis.

**Abstract:**

Acute myeloid leukemia (AML) is a heterogeneous disease. A significant proportion of AML patients is refractory to clinical treatment or relapses. Our aim is to determine new potential AML clinical treatment prognosis markers. We investigated various cell fate and epigenetic regulation important gene level differences between refractory and responsive AML patient groups at diagnosis stage and after clinical treatment using RT-qPCR. We demonstrated that oncogenic *MYC* and *WT1* and metabolic *IDH1* gene expression was significantly higher and cell cycle inhibitor *CDKN1A (p21)* gene expression was significantly lower in refractory patients’ bone marrow cells compared to treatment responsive patients both at diagnosis and after clinical treatment. Moreover, we determined that, compared to clinical treatment responsive patients, refractory patients possess a significantly higher gene expression of histone deacetylase 2 (*HDAC2*) and epigenetic DNA modulator *TET1* and a significantly lower gene expression of lysine acetyltransferase 6A (*KAT6A*) and nucleosome remodeling and deacetylase (NuRD) complex component *GATAD2A*. We suggest that *MYC*, *WT1*, *IDH1*, *CDKN1A*, *HDAC2*, *TET1*, *KAT6A* and *GATAD2A* gene expression changes might characterize refractory AML. Thus, they might be useful for AML prognosis. Additionally, we suggest that epigenetic modulation might be beneficial in combination with standard treatment.

## 1. Introduction

Acute myeloid leukemia (AML) is characterized by the uncontrolled proliferation of neoplastic myeloid cells, which leads to the disruption of hematopoiesis [1]. This disease is very heterogeneous: its cytogenetic heterogeneity was identified decades ago; however, the scale and prognostic importance of its molecular heterogeneity has become evident only during the last two decades [2]. Although karyotype is the strongest prognostic parameter in AML patients, approximately 45% of AML patients feature a normal karyotype [3]. Thus, next-generation sequencing technologies were applied to detect somatic mutations, which help to predict AML risk [4]. The mutations affecting *FLT3*, *NPM1*, *CEBPA* and *RUNX1* genes are among the most clinically important mutations in AML [3]. Epigenetic dysregulation also plays an important role in AML. It was demonstrated that a great proportion of AML cases had mutations in epigenetic regulators, of which 44 % were detected in DNA methylation-related genes and 43% in chromatin modifier or cohesin-complex genes [4].

Treatment outcome is influenced not only by genetic profile, but also by age. Another important prognostic factor is patient ability to respond to intensive induction chemotherapy—failure to respond estimates poor prognosis [5]. Conventional chemotherapy 7 + 3 (7 days of cytarabine + 3 days of anthracycline) cures approximately 40% of younger and 15% of older adult AML patients. The survival rate for patients who are refractory to the treatment or relapse is only approximately 10% [6]. For intermediate and poor prognosis patients who manage to achieve remission, hematopoietic stem cell transplantation (HSCT) is often used; however, the long-term survival rate remains low [7]. 

Thus, AML is still a challenge: not only new treatment approaches, but also better understanding of this disorder, are necessary. In this study, we split AML patients into two groups depending on their response to the clinical treatment and compared the expression levels of the tested genes between them. After a thorough literature analysis, we chose the genes that have been shown to have an important role in AML pathogenesis. Therefore, we chose to analyze cell cycle, apoptosis, metabolism and epigenetic-regulation-related genes, such as genes that tend to be deregulated in cancerous cells. 

## 2. Materials and Methods

### 2.1. Patients

Bone marrow samples were obtained by aspiration from treatment-responsive (*n* = 15) and treatment-resistant (*n* = 16) adult AML patients at diagnosis and after treatment (their diagnosis and treatment are provided in Appendix A). Mononuclear cells were purified using Ficoll-Paque PLUS density gradient centrifugation (GE Healthcare Chicago, IL, USA). The study was conducted in accordance with the Declaration of Helsinki, and the protocol was approved by the Vilnius Regional Biomedical Research Ethics Committee (Approval No. 158200-16-824-356). Informed consent was obtained from all individual participants included in the study.

### 2.2. Gene Expression Analysis by RT-qPCR

Total RNA was purified using RNeasy Mini Kit (QIAGEN, Hilden, Germany), cDNA was synthesized using SensiFAST™ cDNA Synthesis Kit (Bioline, Memphis, TN, USA) and qPCR was performed using SensiFAST™ SYBR^®^ No-ROX Kit (Bioline) on the RotorGene 6000 system (Corbett Life Science, QIAGEN) according to manufacturers’ instructions. Primer sequences (Metabion international AG, Planegg/Steinkirchen, Germany) are outlined in Appendix A. Relative gene expression was calculated using ∆∆Ct method. mRNA levels were normalized to *GAPDH* expression. 

### 2.3. Statistical Analysis

Results are presented as mean ± standard deviation (S.D.); grey data points indicate outliers. Mann–Whitney U test was used to determine the significance of difference between groups of different patients’ samples, and significance was set at *p* ≤ 0.05 (*). Outliers were determined by ROUT (Q = 5%). Statistical analysis was performed using GraphPad Prism version 8.0.1 for Windows (GraphPad Software, San Diego, CA, USA).

## 3. Results

### 3.1. Cell-Fate-Important Gene Expression in AML Patients

AML patients were split into two groups: the ones who responded to clinical treatment and reached remission (*n* = 15) and the ones who showed resistance to the used clinical treatment (*n* = 16). The samples were collected at diagnosis stage and after the clinical treatment. Mononuclear cells were purified and used for gene expression analysis by RT-qPCR. We analyzed some cell-fate-important gene expression differences between treatment-resistant and treatment-responsive patients at diagnosis and after clinical treatment. The expression of the oncogenes *MYC* and *WT1* was significantly higher in refractory patients at diagnosis compared to responsive patients. Moreover, it significantly decreased after successful treatment, while remaining unchanged in refractory patients (Figure 1). Isocitrate dehydrogenase 1 coding gene (*IDH1*) expression was also demonstrated to be significantly higher in refractory AML patients. Meanwhile, cell-cycle-inhibitor *CDKN1A (p21)* gene expression was significantly higher in the treatment-responsive patient group before and after the treatment. Interestingly, pro-apoptotic gene *DAPK1* expression was demonstrated to be significantly higher, while anti-apoptotic *BCL2A1* expression showed a tendency to be lower in refractory patients (Figure 1). We did not find any significant differences between the tested patient groups in the gene expression of the anti-apoptotic genes *BCL2*, *BCL2L1*, *BCL2L2* and *MCL1*; the pro-apoptotic genes *APAF1*, *BAK1*, *BAX* and *P53*; and also some other cell-fate-important genes, including *TGFBR1*, *ABCB1*, *BECN1* and *IDH2* (Figure 1). In brief, the presented results show that refractory AML patients demonstrate significantly higher oncogenic *MYC* and *WT1*, metabolic *IDH1* and pro-apoptotic *DAPK1* gene expression and significantly lower cell-cycle-inhibitor *CDKN1A (p21)* gene expression both at diagnosis and after clinical treatment. 

### 3.2. Chromatin-Remodeling-Related Gene Expression in AML Patients

We also revealed some significant differences in epigenetic-regulation-related gene expression between treatment-refractory and treatment-responsive patients. Histone deacetylase 2 is a class I deacetylase coded by gene *HDAC2*; it causes transcriptional repression by acetyl group removal from histone-specific lysine residues [8]. GATA Zinc Finger Domain Containing 2A (GATAD2A, also known as p66α) is a subunit of the nucleosome remodeling and deacetylase (NuRD) complex. NuRD is highly conserved chromatin-remodeling complex that participates in DNA-damage-induced transcriptional repression [9]. The chromatin remodeling complex NuRD role in tumorigenesis was demonstrated to depend on context: it might either promote or suppress tumorigenesis [10]. SIN3A represses gene transcription by recruiting histone deacetylases to modify chromatin at particular sites of the genome [11]. We evaluated gene expression differences of these transcription repressors in AML patients and determined that *HDAC2* gene expression was significantly higher, while *GATAD2A* was significantly lower in refractory AML patients compared to clinical-treatment-responsive patients. Additionally, in refractory AML patients, *SIN3A* gene expression was lower at diagnosis compared to responsive patients (Figure 2A).

We also demonstrated differences between AML patient groups in transcription-activating histone modificators. Lysine acetyltransferase 6A (KAT6A, also known as MOZ or MYST3) is a histone acetyltransferase required for H3K9 acetylation at target loci [1]. It participates in the regulation of various fundamental cellular processes, including cell cycle progression and stem cell maintenance [12,13,14]. In our study, *KAT6A* gene expression was significantly lower in refractory AML patients compared to responsive patients (Figure 2B). Lysine demethylase 6B (KDM6B, also known as JMJD3) is a H3K27me3-specific demethylase. It acts as a competitor of the polycomb-repressive complex 2 (PRC2), which adds methyl groups to H3K27. Thus, KDM6B activates gene transcription [15]. We showed that *KDM6B* gene expression had a tendency to be lower in refractory AML patients compared to the responsive patient group. Lysine acetyltransferase 2A (KAT2A) enhance transcriptional activity through histone and other target protein acetylation [16,17]. In our study, *KAT2A* gene expression showed a tendency to be higher in patients that are refractory to the treatment. We also evaluated *EZH2*, *HDAC1*, *MTA1* and *MTA2* and *KAT2B* gene expression, but they did not show any significant differences between tested groups (Appendix A). In short, the histone-modification-related gene expression analysis revealed that refractory AML patients are characterized by significantly higher *HDAC2* and significantly lower *GATAD2A* and *KAT6A* expression. Additionally, *SIN3A* gene expression was significantly lower at refractory patient diagnosis.

### 3.3. Epigenetic DNA Modification-Related Gene Expression in AML Patients

DNA methylation status also has an important role in cancer progression. Thus, we investigated the gene expression of several players participating in DNA methylation. TET1 is a dioxygenase that catalyzes 5-mC conversion to 5-hmC (5-hydroxymethylcytosine) and, therefore, participates in the DNA demethylation process [18]. Meanwhile, DNMT1 and DNMT3A are DNA methyltransferases that catalyze DNA methylation. DNMT3A catalyzes de novo DNA methylation, while DNMT1 preserves DNA methylation patterns during cell division [19]. In the present study, we determined that *TET1* gene expression was significantly higher in refractory AML patients compared to treatment-responsive AML patients. Similarly, *DNMT1* gene expression level was significantly higher at diagnosis in refractory patients. *DNMT3A* gene expression shows a slight tendency to be upregulated in refractory patients (Figure 3). *TET2* and *TET3* expression analysis did not show any significant differences between the tested groups (Appendix A). 

To conclude our obtained results, all detected gene expression differences are outlined in Figure 4 by presenting the proposed refractory AML phenotype.

## 4. Discussion

In the study, we analyzed gene level differences between clinical-treatment-responsive and -refractory AML patients at diagnosis and after the treatment. We revealed that oncogenic *MYC* and *WT1* and metabolic *IDH1* expression was significantly higher in the refractory patient group. *WT1* and *MYC* overexpression is known to be associated with carcinogenesis, thus they are called oncogenes. *WT1* is overexpressed in approximately 90% of AML patients and it is already being suggested as possible prognostic marker for relapse prediction [20]. Similarly, *MYC* activation is indicated as a molecular hallmark of cancer [21]. Its overexpression has also been linked with AML [22]. *IDH1* is recognized to be mutated in 7–14% of AML. There is even a drug approved by the FDA to inhibit mutated IDH1 [23]. Moreover, a high *IDH1* expression was demonstrated to be associated with poor cytogenetically normal AML prognosis [24]. Thus, our findings comply with previously shown data. Our results do not reveal any correlation between classical AML genes, such as *TP53*, *TET2*, *IDH2*, and *EZH2*, expression and patient response to the treatment. These genes are well known to be mutated in AML. For example, TP53 is a tumor suppressor gene; thus, its loss-of-function mutation is associated with the poor survival of AML [25]. Since we did not detect any significant differences between responsive and refractory patients’ *TP53*, *TET2*, *IDH2*, *EZH2* expression, we suggest that the expression of these classically mutated AML genes is not a feature of AML and could not be used for its prognosis. Furthermore, two patients in our resistant patients’ group had a *TP53* mutation, but, in our study, mutated *TP53* expression was very similar to unmutated *TP53* gene expression. We demonstrated that *CDKN1A (p21)* gene expression was significantly lower in refractory patients both at diagnosis and after clinical treatment. Since p21 is a cell cycle inhibitor, it is responsible for cell growth and proliferation control [26]. It was demonstrated that increased *CDKN1A* expression and decreased *MYC* expression contribute to the strong antileukemic effects of the transcription factor KLF4 [27]. Thus, the lower expression of *p21* might be a marker of poor AML treatment prognosis. A dysregulated apoptosis mechanism is among the most important mechanisms used by cancerous cells. There are several pharmaceutical agents targeting the apoptosis pathway developed for AML treatment [28]. Interestingly, according to our results, resistance to apoptosis does not appear to be the main reason of AML resistance to clinical treatment. We tested various apoptosis-related gene expressions (Appendix A) and significant differences were detected only in pro-apoptotic *DAPK1*. Oppositely to what was expected, pro-apoptotic *DAPK1* expression was higher and anti-apoptotic *BCL2A1* expression showed a tendency to be lower in refractory patients (Figure 1). It indicates that some treatment-refractory AML patients might even activate apoptosis induction mechanisms to overcome factors that cause cell resistance to clinical treatment.

Since epigenetic changes participate in the pathogenesis of AML, we compared the epigenetic-regulation-related gene expression between refractory and responsive patient groups. For example, various tumors demonstrate aberrant HDAC expression. It is well known that the inhibition of HDACs leads to anti-cancerous changes, such as decreased cell proliferation and induced apoptosis. Therefore, HDAC inhibitors are being developed for cancer, including AML, treatment. Although HDAC inhibitors are not sufficient as a monotherapy, they show promising results while used in combination with standard therapeutics [29]. The gene expression profiling of AML cells after HDAC1/2 selective inhibition alone and after HDAC1/2 selective inhibition together with treatment with azacytidine relieved particular transcription and cell-cycle-regulation-related genes that might be responsible for the mediation of combinatorial effect [30]. Moreover, it was demonstrated in some cancers that higher HDAC expression might be associated with poor prognosis—for example, in prostate carcinoma, a higher *HDAC2* expression is associated with shorter relapse time [31]. We evaluated *HDAC1* (Appendix A) and *HDAC2* (Figure 2A) expression in both treatment-resistant and -responsive AML patient groups and revealed that *HDAC2* expression was significantly increased in refractory group. We also demonstrated that two other our tested transcription repressors *SIN3A* and *GATAD2A* showed significant differences between treatment-refractory and -responsive AML patient groups: their expression was lower in the refractory group. It was demonstrated that the lower expression of *SIN3A* was associated with more aggressive lung cancer progression [32]. Additionally, SIN3A knockdown increased the metastatic potential of breast cancer cells [11]. GATAD2A can have either cancer suppressive or cancer promotive role depending on the context since it is a nucleosome remodeling and deacetylase (NuRD) complex subunit. NuRD complex function depends largely on its interaction with other proteins [10]. Thus, more comprehensive studies are required to better understand GATAD2A role in AML tumorigenesis and treatment.

Transcription activating histone modificators also play an important role in carcinogenesis. We demonstrated that histone acetyltransferase *KAT6A* gene expression was significantly lower in the refractory patient group. This enzyme was shown to be often involved in carcinogenesis. Chromosome rearrangements resulting in fusion proteins, such as KAT6A-CBP or KAT6A-TIF2, have been identified in AML [13]. In addition, the overexpression of *KAT6A* was demonstrated to enhance PI3K/AKT signaling and tumorigenesis in glioblastoma cells [14]. Thus, KAT6A inhibitors are being developed for cancer therapy [33]. However, this histone acetyltransferase participates in the differentiation of erythroid and myeloid cells. KAT6A interacts with AML1 and acts as a transcriptional coactivator, but fusion protein KAT6A-CBP inhibits AML1-mediated transcription [34]. Thus, KAT6A role in tumorigenesis might depend on the type of cancer and on the presence of aberration. Moreover, it was demonstrated that KAT6A is essential for the ability of hematopoietic stem cells to reconstitute the hematopoietic system of a recipient after transplantation [35]. Our findings that *KAT6A* expression was higher in the treatment-responsive patient group prove its importance for healthy haematopoiesis.

Similarly, lysine demethylase 6B (KDM6B) has a complex role in tumorigenesis. Our results show that *KDM6B* gene expression had a tendency to be downregulated in the refractory AML patient group. However, this gene overexpression was detected in various blood disorders, including myelodysplastic syndromes (MDS), Hodgkin’s lymphoma (HL), multiple myeloma (MM) and T-cell acute lymphoblastic leukemia (T-ALL), implying the need to inhibit KDM6B for successful cancer therapy [36,37,38]. Furthermore, there are ongoing studies working on the development of KDM6B inhibitors for the treatment of hematopoietic disorders, including AML [39]. In contrast, other investigators demonstrated that KDM6B relieves the differentiation arrest of certain subtypes of AML cells. The authors determined that this histone demethylase activates the expression of a number of key myelopoietic regulatory genes by directly modulating H3K4 and H3K27 methylation levels, which indicates KDM6B onco-repressive activity. They determined that KDM6B downregulation correlates with poor clinical outcomes in certain AML subtypes [15]. Thus, KDM6B role in leukemogenesis might depend on the phase and linage of the disease [15,36].

Meanwhile, our tested transcription activating modificatory lysine acetyltransferase 2A (KAT2A) showed a tendency to be upregulated in refractory AML patients. This enzyme also plays different roles in different cancer types. For example, it was revealed that high *KAT2A* expression is associated with poor prognosis in breast, lung and colon cancers; however, it is associated with good prognosis in pancreatic adenocarcinoma and glioma. Its oncogenic role might be explained by causing the histone acetylation-mediated activation of E2F and MYC targets that participate in maintaining cell proliferation and survival. In AML, it was demonstrated that KAT2A helps to maintain undifferentiated leukemic cells. Additionally, KAT2A knock-out in leukemia stem-like cells lost repopulating capacity [40]. Thus, our findings that the higher expression of *KAT2A* might predict poor AML prognosis conforms to previous research.

We also determined differences between refractory and responsive AML patient groups in DNA methylation related gene expression—*TET1* and *DNMT1* were up-regulated in the refractory group. One scientist group determined that *TET1* expression was significantly reduced in AML patients [18]. Other scientist group performed RNA expression profiling in AML patients possessing high/low *TET1* expression and revealed that high *TET1* expression predicts poor survival in two tested cytogenetically normal AML cohorts and is associated with a short survival time [41]. Thus, our results conform to the latter group’s results. *DNMT1* and *DNMT3A* expression was shown to be increased in AML compared to bone marrow cells from healthy donors previously [42]. Additionally, *DNMT1* expression was upregulated in multi-drug resistant HL60/ATRA cells in vitro [43]. Our results in ex vivo studies also indicate that an elevated *DNMT1* expression might be a sign of AML resistance to treatment. One of the possible DNMT1 oncogenicity explanations is that it causes tumor suppressor p15 downregulation [42].

## 5. Conclusions

In conclusion, we evaluated gene expression differences between refractory and responsive AML patient groups and determined that refractory AML patients showed significantly higher oncogenic *MYC* and *WT1*, metabolic *IDH1* and pro-apoptotic *DAPK1* gene expression and significantly lower cell-cycle-inhibitor *CDKN1A (p21)* gene expression both at diagnosis and after clinical treatment. Moreover, we demonstrated significant differences in epigenetic-regulation-related gene expression: *HDAC2* and *TET1* gene expression was significantly higher, while *GATAD2A* and *KAT6A* was significantly lower in refractory AML patients compared to the responsive patients’ group (Figure 4). Our results reveal that the epigenetic landscape plays an important role in maintaining refractory AML. Our detected gene expression changes might be useful for AML prognosis. Additionally, we suggest that epigenetic modulators, when used in combination with standard treatment, could be useful for refractory AML clinical treatment.

## Figures and Tables

**Figure 1 cancers-14-02752-f001:**
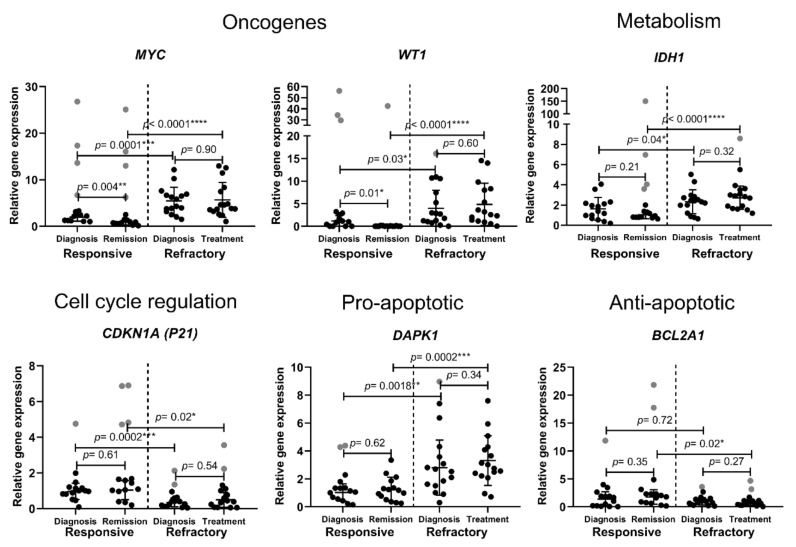
Cell-fate-important gene expression analysis in treatment-responsive and -refractory AML patients. Cell samples were collected at diagnosis stage and after treatment. Relative gene expression analysis was performed using the RT-qPCR ΔΔCt method; *GAPDH* was used as a “housekeeping” gene. Mean ± standard deviation is presented; grey data points indicate outliers. Mann–Whitney U test was used to determine the significance of difference between the groups of different patients’ samples, and significance was set at *p* ≤ 0.05 (*). Outliers were determined by ROUT (Q = 5%).

**Figure 2 cancers-14-02752-f002:**
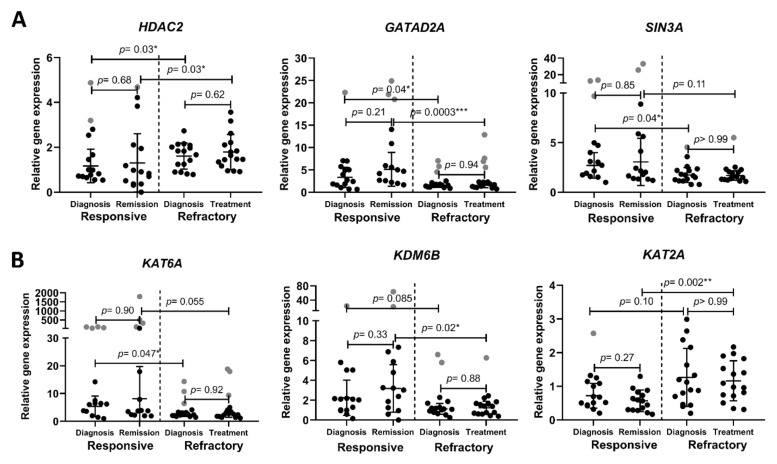
Chromatin-remodeling-related gene expression analysis in treatment-responsive and -refractory AML patients. (**A**) Gene expression changes of transcription-repressing histone modificators; (**B**) gene expression changes of transcription-activating histone modificators. Cell samples were collected at diagnosis stage and after treatment. Relative gene expression analysis was performed using the RT-qPCR ΔΔCt method; *GAPDH* was used as a “housekeeping” gene. Mean ± standard deviation is presented; grey data points indicate outliers. Mann–Whitney U test was used to determine the significance of difference between groups of different patients’ samples, and significance was set at *p* ≤ 0.05 (*). Outliers were determined by ROUT (Q = 5%).

**Figure 3 cancers-14-02752-f003:**
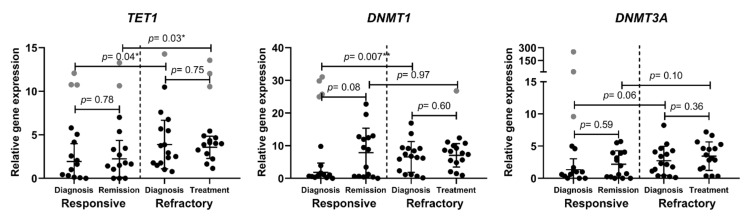
Epigenetic DNA modification-related gene expression analysis in treatment-responsive and -refractory AML patients. Cell samples were collected at diagnosis stage and after treatment. Relative gene expression analysis was performed the using RT-qPCR ΔΔCt method; *GAPDH* was used as a “housekeeping” gene. Mean ± standard deviation is presented; grey data points indicate outliers. Mann–Whitney U test was used to determine the significance of difference between groups of different patients’ samples, and significance was set at *p* ≤ 0.05 (*). Outliers were determined by ROUT (Q = 5%).

**Figure 4 cancers-14-02752-f004:**
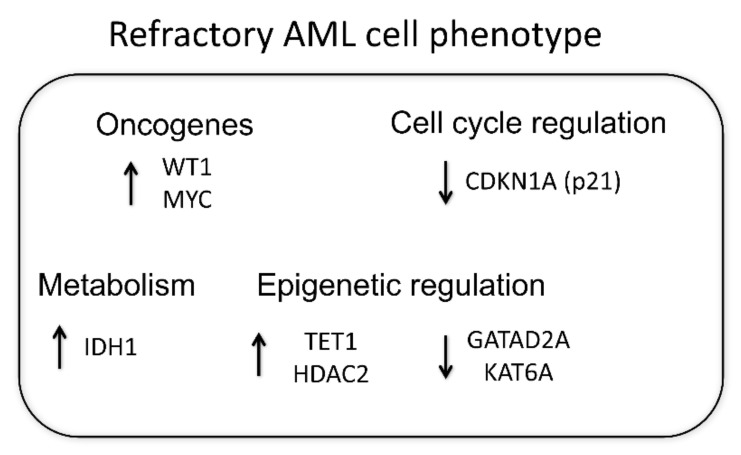
Proposed refractory AML phenotype. AML patients were split into two groups depending on their response to the clinical treatment. Relative gene expression analysis was performed using the RT-qPCR ΔΔCt method; *GAPDH* was used as a “housekeeping” gene. The determined statistically significant gene expression differences between refractory and responsive patients are presented.

## Data Availability

The data presented in this study are available upon request from the corresponding author.

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
