# Peer review of "Potential Prognostic Markers for Relapsed/Refractory vs. Responsive Acute Myeloid Leukemia"

_cancers, 2022, doi:10.3390/cancers14112752_

Round 1

Reviewer 1 Report

The manuscript titled “Potential Prognostic Markers for Relapsed/Refractory vs. Responsive Acute Myeloid Leukemia” addressed certain genes in therapy-resistant and responsive AML patients at diagnosis stage and after clinical treatment. In the research, bone marrow samples from 31 adult AML patients were collected and divided in two cohorts: responsive and resistance groups. Then certain genes RNA expression levels, including cell fate genes, chromatin remodelling related genes and epigenetic genes, were qualified by qPCR. Finally, authors found MYC, WT1, IDH1, CDKN1A, HDAC2, TET1, and etc genes might changes in refractory AML. Overall, the whole story is a straightforward and the article is well organized. However, some issues still need to be improved:

  1. Some classic genes like p53 and TET2, ASXL1 (Gu, R., Yang, X. & Wei, H. Molecular landscape and targeted therapy of acute myeloid leukemia. Biomark Res 6, 32 (2018)) should be discussed why not related to the refractory.
  2. It could be much better if authors could combine large scale data analysis from platform to verify certain genes.
  3. Besides critical genes expression, patients’ other factors like age, Karyotype also influence prognose and survival. Authors may take these factors into consideration and comprehensively analysis.

Author Response

Q: Some classic genes like p53 and TET2, ASXL1 (Gu, R., Yang, X. & Wei, H. Molecular landscape and targeted therapy of acute myeloid leukemia. Biomark Res 6, 32 (2018)) should be discussed why not related to the refractory.

A: Thank you for the remark. We expanded our discussion in lanes 207-213: “Our results did not reveal any correlation between classical AML genes’, such as TP53, TET2, IDH2, EZH2, expression and patient response to the treatment. These genes are well known to be mutated in AML. For example, TP53 is a tumor suppressor gene, thus, its loss-of-function mutation is associated with poor survival of AML [25]. Since we did not detect any significant differences between responsive and refractory patients’ TP53, TET2, IDH2, EZH2 expression, we suggest that the expression of these classically mutated AML genes is not a feature of AML and could not be used for its prognosis.”

Q: It could be much better if authors could combine large scale data analysis from platform to verify certain genes.

A: As the reviewer suggested, we analyzed several expression databases looking for validation of our results. However, it is difficult to compare the results between different experiments as patients are grouped using different factors. Also, in many databases, less emphasis is placed on gene amplification than on mutations. It is important to mention that in our study we grouped patients according to their response to the treatment, not according to their ELN risk group.  Also, we analyzed gene expression of the same patient at diagnosis and again at follow-up stage (after medical treatment).

Q: Besides critical genes expression, patients’ other factors like age, Karyotype also influence prognose and survival. Authors may take these factors into consideration and comprehensively analysis.

A: We agree with the reviewer that such factors as age and karyotype are very important for AML prognosis. These criteria are already used in hospitals to decide patient’s risk group and we added this information in Supplementary Table 1. Since presently known prognostic factors are not sufficient to decide precise prognosis and treatment options, we aimed to find further possible prognostic markers which could not only suggest more precise prognosis but also could suggest treatment possibilities.

We hope that we have addressed all the concerns and comments raised. We would like to thank the Reviewer for the time and efforts spent on our manuscript.

Reviewer 2 Report

The manuscript from  Vitkevičienė A. et al,  collected samples from AML patients at diagnosis and after treatment and divided them into responsive and refractory (16-16 patients per group).  They attempted to understand if there are any biomarkers that will help to predict the treatment outcome.

In order to do that, they tested the expression of a set of genes and correlated their expression patters to the outcome.

It is a very straight forward experiment, and some of the results are interesting, but in my view preliminary.

There are many studies in AML patients with genome wide and unbiased analysis (RNA-seq/microarrays and even scRNA-seq, some of the data accessible via TCGA cohort/GEO NCBI or https://ihec-epigenomes.org/) that look at similar idea, the most recent being PMID 35241089, PMID: 33854980. The contribution of this work is therefore very limited.

I have some issues with the design:

  1. The set of genes they decided to study are well knows genes involved in AML pathogenesis, diagnosis and prognosis, so it is not a surprise that some of them seem to have a predictive value.
  2. How did the authors decided the set of genes to test?
  3. There is no cross validation of their results with the several expression datasets?
  4. They did not do any other analysis to see is there is any other correlation. The treatment is not the same in all cases.
  5. DO the patients they excluded from the analysis are the same in all plots? If that is the case, what could be the cause for the ‘abnormal’ expressions patterns?
  6. They claimed to have used the ΔΔCt method, but as far as I can see they did ΔCt method to GAPDH expression, because they are normalizing against diagnosis
  7. The discussion is very limited
  8. How do the molecular genetic (mutational) burden of the patients (in the same genes tested) affected their expression?

Minor: The figures are too small

Information of the primers sets are not given

Author Response

1. Q: The set of genes they decided to study are well knows genes involved in AML pathogenesis, diagnosis, and prognosis, so it is not a surprise that some of them seem to have a predictive value.

A: We agree that substantial part of tested genes is known to be involved in AML. We would like to emphasize that usually mutated versions of those genes are said to be important and used for AML diagnosis and prognosis, but here we tested the possible importance of their expression.

2. Q: How did the authors decided the set of genes to test?

A: The set of genes was selected after thorough literature analysis. Genetic abnormalities of AML promote leukemic stem cell formation by interfering with hematopoietic differentiation and enhancing the self-renewal capacity of hematopoietic cells. Gene mutations recurrently occur in transcription factors, signaling molecules, tumor suppressor genes, epigenetic regulators, RNA splicing factors, and cohesion complexes. Since AML disrupts various cellular processes, we chose to test the expression of genes related to cell cycle regulation, autophagy, apoptosis, epigenetic regulation, and oncogenes. We clarified this point in the manuscript in lanes 61-64: “After thorough literature analysis, we chose the genes that have been shown to have an important role in AML pathogenesis. Therefore, we chose to analyze cell cycle, apoptosis, metabolism and epigenetic regulation related genes as such genes tend to be deregulated in cancerous cells.”

3. Q: There is no cross validation of their results with the several expression datasets?

A: As the reviewer suggested, we analyzed several expression databases looking for validation of our results. However, it is difficult to compare the results between different experiments as patients are grouped using different factors. It is important to mention that in our study we grouped patients according to their response to the treatment, not according to their ELN risk group.  Also, we analyzed gene expression of the same patient at diagnosis and again at follow-up stage (after medical treatment). The latter helped to prove the importance of found differences in gene expression at diagnosis. Using the proposed databases, we found some related work which we used to expand the discussion of the manuscript: Lanes 218-220: “It was demonstrated that increased CDKN1A expression and decreased MYC expression contribute to strong antileukemic effects of transcription factor KLF4”. Lanes 239-242: Gene expression profiling of AML cells after HDAC1/2 selective inhibition alone and after HDAC1/2 selective inhibition together with treatment with azacytidine relieved particular transcription and cells cycle regulation related genes that might be responsible for mediation of combinatorial effect. Lanes 301-304: “Other scientist group performed RNA expression profiling in AML patients possessing high/low TET1 expression and revealed that high TET1 expression predicts poor survival in two tested cytogenetically normal AML cohorts and is associated with short survival time”.

4. Q: They did not do any other analysis to see is there is any other correlation. The treatment is not the same in all cases.

A: Since every patient demands different clinical approach, indeed, the treatment is not the same in all cases. We tried to perform analysis based on genetic mutations/karyotype, however, we faced the issue of sample size. As AML patients have very heterogenous karyotypes and mutations, we could not group them sufficiently to reveal significant correlation to gene expression patterns and obtain reliable results.

5. Q: DO the patients they excluded from the analysis are the same in all plots? If that is the case, what could be the cause for the ‘abnormal’ expressions patterns?

A: Patients excluded from the analysis are not the same in all plots. Exclusion was performed with GraphPad Prism, only mathematically, using Rout outliers’ detection method with Q=5. The ROUT method is based on the False Discovery Rate (FDR). Q is the maximum desired FDR. The interpretation of Q depends on whether there are any outliers in the data set. When there are no outliers (and the distribution is entirely Gaussian), Q is very similar to alpha. When there are outliers in the data, Q is the maximum desired false discovery rate. If Q is set to 5%, then no more than 5% of the identified outliers to be false.

6. Q: They claimed to have used the ΔΔCt method, but as far as I can see they did ΔCt method to GAPDH expression, because they are normalizing against diagnosis

A: Thank you for the remark. Our calculation workflow was as follows: to begin with, we performed RT-qPCR of reference gene GAPDH and of target genes; then, we subtracted the average Ct value of GAPDH from target gene to obtain ΔCt; later, we subtracted ΔCt of target sample from ΔCt of reference sample (one of tested patients diagnosis sample value) to obtain ΔΔCt – relative gene expression value.

7. Q: The discussion is very limited

A: We expanded and improved the discussion section of the manuscript. Lanes 207-215: Our results did not reveal any correlation between classical AML genes’, such as TP53, TET2, IDH2, EZH2, expression and patient response to the treatment. These genes are well known to be mutated in AML. For example, TP53 is a tumor suppressor gene, thus, its loss-of-function mutation is associated with poor survival of AML [25]. Since we did not detect any significant differences between responsive and refractory patients’ TP53, TET2, IDH2, EZH2 expression, we suggest that the expression of these classically mutated AML genes is not a feature of AML and could not be used for its prognosis. Furthermore, two patients in our resistant patients’ group had TP53 mutation but, in our study, mutated TP53 expression was very similar to unmutated TP53 gene expression. Also already mentioned in the answer no.3: Lanes 218-220: “It was demonstrated that increased CDKN1A expression and decreased MYC expression contribute to strong antileukemic effects of transcription factor KLF4”. Lanes 239-242: Gene expression profiling of AML cells after HDAC1/2 selective inhibition alone and after HDAC1/2 selective inhibition together with treatment with azacytidine relieved particular transcription and cells cycle regulation related genes that might be responsible for mediation of combinatorial effect. Lanes 301-304: “Other scientist group performed RNA expression profiling in AML patients possessing high/low TET1 expression and revealed that high TET1 expression predicts poor survival in two tested cytogenetically normal AML cohorts and is associated with short survival time”.

8. Q: How do the molecular genetic (mutational) burden of the patients (in the same genes tested) affected their expression?

A: In this study, we analyzed the expression of TP53 gene, which is frequently mutated in AML and was mutated in two tested patients (both in refractory group). There was no significant difference in expression of TP53 gene in Responsive vs. Refractory groups nor in Diagnosis vs. Treatment. The expression of two mutated TP53 samples was very similar to unmutated TP53 gene expression. We expanded the discussion with this important remark in lanes 213-215: “Furthermore, two patients in our resistant patients’ group had TP53 mutation but, in our study, mutated TP53 expression was very similar to unmutated TP53 gene expression.”

Q: Minor: The figures are too small

A: We improved the quality and size of the figures.

Q: Information of the primers sets are not given

A: We included primer sequences in Supplementary Table 2.

We hope that we have addressed all the concerns and comments raised. We would like to thank the Reviewer for the time and efforts spent on our manuscript.

Reviewer 3 Report

In "Potential Prognostic Markers for Relapsed/Refractory vs. Responsive Acute Myeloid Leukemia" Aida Vitkevičienė and coll well describe that MYC, 
WT1, IDH1, CDKN1A, HDAC2, TET1, KAT6A and GATAD2A gene expression changes might characterize refractory AML. Thus, they might be useful for AML prognosis. At the same time they suggest as  epigenetic modulation might be beneficial in combination with standard treatment. In real life refractory AMLs receive this therapeutic option, as bridge to allogeneic transplant. Even though the gene expression analysis is only performed by RT-qPCR, results are clear and reliable. The study design is good and also the supplement data about patient's clinical features are sufficient. 

It's a good job, not so original or innovative, but fine

Author Response

Q: In "Potential Prognostic Markers for Relapsed/Refractory vs. Responsive Acute Myeloid Leukemia" Aida Vitkevičienė and coll well describe that MYC, WT1, IDH1, CDKN1A, HDAC2, TET1, KAT6A and GATAD2A gene expression changes might characterize refractory AML. Thus, they might be useful for AML prognosis. At the same time, they suggest as epigenetic modulation might be beneficial in combination with standard treatment. In real life refractory AMLs receive this therapeutic option, as bridge to allogeneic transplant. Even though the gene expression analysis is only performed by RT-qPCR, results are clear and reliable. The study design is good and also the supplement data about patient's clinical features are sufficient. It's a good job, not so original or innovative, but fine.

A: Thank you for your opinion. We are glad that you found our study interesting and useful.

We would like to thank the Reviewer for the time and efforts spent on our manuscript.

Round 2

Reviewer 1 Report

All the questions are answered. I do recommend this manuscript for publication in the journal.